# Natural Compounds for Bone Remodeling: A Computational and Experimental Approach Targeting Bone Metabolism-Related Proteins

**DOI:** 10.3390/ijms25095047

**Published:** 2024-05-06

**Authors:** Alexandros-Timotheos Loukas, Michail Papadourakis, Vasilis Panagiotopoulos, Apostolia Zarmpala, Eleni Chontzopoulou, Stephanos Christodoulou, Theodora Katsila, Panagiotis Zoumpoulakis, Minos-Timotheos Matsoukas

**Affiliations:** 1Department of Food Science and Technology, University of West Attica, Ag. Spyridonos, 12243 Egaleo, Greece; atloukas@uniwa.gr (A.-T.L.); pzoump@uniwa.gr (P.Z.); 2Cloudpharm Private Company, Kifissias Avenue 44, 15125 Marousi, Greece; vpanagiotopoulos@cloudpharm.eu (V.P.); azarmpala@cloudpharm.eu (A.Z.); echontzopoulou@cloudpharm.eu (E.C.); stchristodoulou1997@gmail.com (S.C.); 3Department of Biomedical Engineering, University of West Attica, Ag. Spyridonos, 12243 Egaleo, Greece; 4Institute of Chemical Biology, National Hellenic Research Foundation, 11635 Athens, Greece; thkatsila@eie.gr

**Keywords:** osteoporosis, natural compounds, molecular docking, molecular dynamics simulations, bone metabolism, quercetin, MAPKs

## Abstract

Osteoporosis, characterized by reduced bone density and increased fracture risk, affects over 200 million people worldwide, predominantly older adults and postmenopausal women. The disruption of the balance between bone-forming osteoblasts and bone-resorbing osteoclasts underlies osteoporosis pathophysiology. Standard treatment includes lifestyle modifications, calcium and vitamin D supplementation and specific drugs that either inhibit osteoclasts or stimulate osteoblasts. However, these treatments have limitations, including side effects and compliance issues. Natural products have emerged as potential osteoporosis therapeutics, but their mechanisms of action remain poorly understood. In this study, we investigate the efficacy of natural compounds in modulating molecular targets relevant to osteoporosis, focusing on the Mitogen-Activated Protein Kinase (MAPK) pathway and the gut microbiome’s influence on bone homeostasis. Using an in silico and in vitro methodology, we have identified quercetin as a promising candidate in modulating MAPK activity, offering a potential therapeutic perspective for osteoporosis treatment.

## 1. Introduction

Bone and mineral metabolism play a crucial role in preserving bone health, while their interruption can cause a decrease in bone density and consequently osteoporosis. Osteoporosis is becoming a serious healthcare issue due to the high morbidity, mortality and significant healthcare cost involved and is characterized by skeletal fragility and susceptibility to fractures [1]. It is a metabolic bone disease that affects over 200 million individuals globally [2], and it is occurring mostly in older adults and postmenopausal women [3]. Two categories of osteoporosis have been identified: primary, when it is age-related, and secondary, when it is caused due to the presence of an underlying disease or mediation [4]. Osteoporosis is caused by the disruption of the cooperative balance between osteoblasts and osteoclasts [5], and it can be diagnosed by measuring the bone mineral density (BMD) with dual X-ray absorptiometry (DXA) [6], which is considered the gold standard for the diagnosis of osteoporosis [7]. BMD (g/cm^2^) is determined by the ratio of the measured bone mineral content to the measured bone area and is converted to the Trabecular Bone Score (T-score). According to the World Health Organization (WHO), osteoporosis occurs when the reference values of the T-score ≤ −2.5 of DXA at the lumbar spine, femoral neck or distal third of the radius [8,9], while a patient is diagnosed with osteopenia when the reference values of the T-score are between −1 and −2.5 [10].

Once osteoporosis has been diagnosed, healthcare professionals can tailor specific interventions to achieve an effective therapy during the early phase of this disease [11]. Fundamental aspects for any osteoporosis prevention or treatment program include lifestyle modifications (physical exercise, balanced diet, the cessation of tobacco use and excess alcohol intake) and vitamin D and calcium supplementation. An adequate intake of calcium decreases the rate of bone mineral density loss [12] and combined with vitamin D reduces the risk of total and hip fractures [13]. Yet, these approaches are insufficient in preventing the progression of osteoporosis, and they need to be combined with specific osteoporosis drugs that have been developed to restore the normal balance between bone resorption and bone formation [14]. Currently available osteoporosis drugs are either antiresorptive, which slow down bone resorption by inhibiting the osteoclasts and anabolic drugs, which increase bone formation by stimulating the osteoblasts [15]. Both therapeutic strategies have demonstrated proven efficacy [16], but they are limited by side effects, concurrent comorbidities and inadequate long-term compliance [17]. Therefore, there is a great need to tailor treatments for osteoporosis that are both effective and safe.

In recent years, natural products have emerged as promising candidates for preventing osteoporosis; however, the exact mechanism through which they operate remains unknown, resulting in limiting clinical applications [18]. In response to this challenge, investigating the potential of natural products to counteract bone fragility based on molecular targets could offer potential therapeutic benefits for patients. Several molecular mechanisms are known to have an impact on bone homeostasis [5,11]. The Wingless-type mouse mammary virus integration site (Wnt) signaling pathway is an evolutionary conserved system of cell–cell communication that is the master switch for osteoblastic differentiation [19]. Sclerostin and dickkopf-1 function as antagonists of Wnt signaling, exerting negative regulatory influence on bone formation [20]. Monoclonal antibodies, such as romosozumab and BHQ 880, have been strategically employed to specifically target and neutralize the inhibitory actions of sclerostin and Dickkopf-1, respectively, thereby facilitating the augmentation of bone formation [21]. Moreover, therapeutic interventions have been implemented to attenuate osteoclastic differentiation which is primarily modulated by the receptor activator of the nuclear factor-kB (RANK)/RANK ligand (RANKL)/osteoprotegerin (OPG) pathway [22]. Denosumab, a monoclonal antibody which successfully neutralizes RANKL and thus inhibits key steps in osteoclast-mediated bone resorption, has clearly shown substantial antifracture benefits in clinical trials [23] and can be considered as a first-line treatment for osteoporosis.

The Mitogen-Activated Protein Kinase (MAPK) pathway can also be a potential therapeutic target for the treatment of osteoporosis. It has been shown to regulate osteoclast differentiation and activation [24] with two Extracellular Signal-Regulated Kinase (ERK) isoforms, ERK1/2, and three c-Jun N-terminal Kinase (JNK) forms, JNK1/2/3, having a primary role in influencing the proliferation of osteoclast precursors as well as the regulation of apoptosis in osteoclasts [25]. Moreover, the four isoforms of p38 (p38α/β/γ/δ) also play a key role in osteoclast differentiation and bone resorption [26]. Therefore, targeting specific components of the MAPK pathway may influence bone remodeling.

Recent studies have also yielded novel insights into the influence of the gut microbiome in bone homeostasis and the mechanism of degenerative skeletal diseases, such as osteoporosis [27,28]. The gut microbiome consists of trillions of symbiotic and pathogenic microorganisms residing in the human intestine that build mutually beneficial relationships with the host [29,30]. It is originally obtained from the mother during birth, and it functions as a multicellular organ which influences a plethora of physiological functions of the human body including metabolic balance [31,32]. In addition, it has been associated with the pathogenesis of several metabolic disorders, including osteoporosis, where an imbalance in the gut microbiome, known as dysbiosis, has been observed in associated patients [33,34,35]. For this reason, it has been considered as an appealing therapeutic strategy for combating osteoporosis.

In pursuit of therapeutic interventions to address osteoporosis, the potential of phytochemicals to modulate the activity of MAPKs related to bone metabolism and osteoporosis was examined against the p38, ERK and JNK isoforms while also investigating their influence on the gut microbiome. A graphical overview of the strategy pursued in this study is depicted in Figure 1. Briefly, several classes of natural compounds were collected from an open access chemical library. The interaction of these molecules with 40 strains of the human gut microbiome that could affect osteoporosis was predicted using a well-established machine learning (ML) algorithm. Virtual screening was performed for the selected protein targets, and the top-ranked compounds were further subjected to 100 ns molecular dynamics (MD) simulations in triplicates to study the structural dynamics, conformational behavior and the stability of protein–ligand complexes. Finally, the binding free energy of each protein–ligand complex was assessed using the Molecular Mechanics Poisson–Boltzmann surface area (MM-PBSA) method to evaluate in vitro selected compounds against specific MAPKs.

## 2. Results and Discussion

### 2.1. The Selection of the Most Promising Natural Compounds for MD Simulations

A combined computational and experimental workflow was implemented for the discovery of the potential inhibitors of specific components of the MAPK pathway that may influence bone remodeling. An overview of this workflow is depicted in Figure 2. To assess the potential of natural compounds to modulate the activity of MAPKs related to osteoporosis, we first selected 187 molecules from an online platform, CNatural, (http://cnatural.eu/home/, accessed on 14 March 2024) based on their reported association in osteoporosis from literature data. To select desirable compounds for MD simulations, four filters were applied: (A) the evaluation of docking scores and pertinent binding poses of the molecules, (B) consideration of the potential issuance of scientific opinions from the European Food Safety Authority (EFSA) and assessing the safety profiles of the 187 natural compounds, (C) examination of their involvement in clinical trials related to osteoporosis and (D) assessment of their influence on gut microbiome strains associated with osteoporosis.

As a first step, the resulting molecules were screened on the selected crystal structures of p38b and ERK1 using the AutoDock Vina 1.1.2 [36] docking software. The choice of crystal structures is described analytically in the Methods section. Of particular interest is the location of the co-crystalized ligands in relation to the ATP-binding site and the type of inhibition that they exhibit. Kinase inhibitors of p38 [37] and JNK [38] isoforms are classified into type-I inhibitors that mimic ATP and aim for the ATP-binding site and type-II inhibitors, which stabilize an “out” conformation of the ATP/Mg^2+^-coordinating domain distinguished by a conserved Asp-Phe-Gly (DFG) motif [39,40]. Finally, ERK isoforms demonstrate minimal tendency towards adopting the ‘DFG-out’ conformation. Therefore, the potent inhibitors target a distinct allosteric pocket located adjacent to the ATP site [41]. The preferred binding modes selected in this study focused on the type-II inhibition for p38 and JNK isoforms along with the allosteric inhibition reported in Chaikuad et al. [41]. The decision was driven by the robust binding of ATP to the ATP-binding site, rendering it less favorable as a primary target for natural compounds. Therefore, all the phytochemicals were docked using AutoDock Vina [36] in order to acquire the lowest energetic structure in the allosteric binding region for each protein. To assess the capability of AutoDock Vina in generating reliable initial structures for MD simulations, we performed the self-docking of the co-crystalized ligands into their respective allosteric binding sites within p38b and ERK1. The ligands consistently attained the highest docking scores while maintaining their positions within the allosteric binding pocket. Specifically, they achieved scores of −11.0 kcal/mol in comparison to −10.5 kcal/mol for the top-ranked compound in ERK1 and −13.7 kcal/mol compared to −11.9 kcal/mol for the top-ranked compound in p38b. The docking scores of the 187 natural compounds on the two proteins are listed in Appendix A.

The compounds were additionally filtered based on their safety profiles. Through comprehensive database searches, legislative inquiries and literature reviews, 40% of the natural compounds were found to have an established daily dose deemed safe and efficacious, demonstrating beneficial effects without any reported adverse side effects. These compounds were further refined based on whether the EFSA had assessed their safety of use in products and their potential beneficial effects on specific functions of the human body. Only 15 compounds have received scientific opinions from the EFSA [42,43,44,45,46,47,48,49,50,51,52,53,54,55,56,57,58], and Appendix A lists the inquiries submitted to the EFSA for the respective phytochemicals along with the corresponding assessments received. Moreover, we explored the possible participation of the 187 natural compounds in clinical trials associated with osteoporosis. Using the clinical.trials.gov database (https://clinicaltrials.gov/, accessed on 14 March 2024), we conducted a search for clinical studies and nutritional interventions involving the 187 natural compounds and their corresponding extracts. Appendix A presents examples of completed clinical studies assessing the impact on bone health, whether through individual phytochemicals, their extracts or diets containing them. The majority of available clinical data pertain to flavonoids (genistein-, hesperidin-, soy isoflavones and isoflavone-enriched foods) and polyphenols (past dried plum, blueberry and green tea extracts) [59,60,61,62,63,64,65,66,67,68,69]. However, due to the non-mandatory nature of conducting clinical studies or nutritional interventions for dietary supplements, insufficient clinical data were identified for the selected compounds or their extracts.

Finally, we examined whether these compounds could inhibit the growth of human gut bacterial strains using an ML model developed by McCoubrey et al. [70]. Among the cohort of 40 strains, 14 strains are affiliated with the genera Ruminococcous, Bifidobacterium, Clostridium, Fusobacterium and Veillonella. According to clinical studies involving patients with osteoporosis and other skeletal health problems, augmented populations of these genera were observed within their human gut microbiome [71,72,73]. Hence, it is crucial to hinder their growth. On the other hand, diminished populations of the genera Bacteroides and Lactobacillus (nine strains found in the dataset) were detected, and it is imperative to avoid inhibiting their growth. Thus, our prioritization of compounds was based on their predicted interaction with both the total number of bacterial strains and the number of strains associated with osteoporosis. These models face limitations primarily due to the availability of data, as the number of studies on microbe–drug interactions remain relatively low, and there is a scarcity of publicly accessible shotgun metagenomic data for analysis. Moreover, technical disparities across studies often hinder comparison and validation efforts. Additionally, the effectiveness of a drug’s anti-commensal effect is contingent upon its bioavailability at the site of action. However, the model utilized relies on data from high-throughput screens conducted at a fixed concentration. While the authors demonstrated that this concentration generally aligns with intestinal levels for most drugs, variations may exist for certain compounds due to influences from both host and microbial factors. Furthermore, the method and analysis here utilize a single strain as a representative of each species, potentially overlooking strain-specific responses to drugs. Such variations may encompass factors known to significantly impact antimicrobial and antidrug resistance, such as the presence of specific xenobiotic-metabolizing enzymes and multidrug resistance transporters [74,75,76]. While high-throughput experiments offer crucial insights into complex biological systems, ML methods can complement these experiments by identifying trends and extrapolating observed patterns. Despite the likelihood of future high-throughput screens providing more valuable information on drug–microbiome interactions, the vastness of this space poses challenges, and experimental methods may struggle to keep pace with the rapid development of new compounds and the discovery and sequencing of new microbial strains. Furthermore, in vitro experiments often require the isolation and culture of assayed microbes, which can be challenging or infeasible, particularly for gastrointestinal and host-associated species. Therefore, we assert that the computational prediction of drug–microbe interactions is not only valuable but also a critical component of future research on the interactions between drugs and the human microbiome, offering numerous applications.

After applying the four aforementioned filters, we narrowed down our selection to five phytochemicals for further assessment using MD simulations. Table 1 presents the five candidate molecules alongside the total number of affected bacterial strains attributed to each compound and the number of strains associated with osteoporosis.

The five selected natural compounds are categorized as either flavonoids (three compounds) or polyphenols (two compounds). Chlorogenic acid ranked 100th for ERK1 and 122nd for p38b, both with a docking score of −8.4 kcal/mol, exhibiting a mere 3.5 kcal/mol distinction from the top-ranked compound of p38b (−11.9 kcal/mol) and a 2.1 kcal/mol variance from the top-ranked phytochemical in ERK1 (−10.5 kcal/mol). It was predicted to impact a total number of 18 out of 40 bacterial strains of the human gut microbiome, with 11 of them linked to osteoporosis. Salidroside, the other phenolic compound chosen for MD simulations, was primarily selected as a negative control for our protocol. Scoring among the lower ranks, it ranked 146th for p38b with −7.7 kcal/mol and 100th for ERK1 with −7.3 kcal/mol. It influenced a modest count of six bacterial strains, with only three of them associated with osteoporosis.

Regarding flavonoids, quercetin achieved moderate scores in both p38b (102nd with −8.3 kcal/mol) and ERK1 (104th with −8.5 kcal/mol), with a slight discrepancy from the top-ranked compounds of p38b (3.6 kcal/mol) and ERK1 (2 kcal/mol). However, it influenced over half of the strains (22) within the human gut microbiome, with 15 of them associated with osteoporosis. Quercetin has captured substantial interest from both medical and research fields due to its promising therapeutic potential. This includes its antioxidant, antitumor, antiatherogenic, antithrombotic and cardioprotective properties [77]. As a result, quercetin has emerged as a prominent bioactive ingredient with promising applications in numerous functional foods and pharmaceutical products. Nonetheless, the limited water solubility of this phytochemical significantly hampers its bioavailability and absorption following oral administration [78]. To address this challenge, quercetin has been incorporated into innovative lipid-based systems to enhance its bioactivity and therapeutic profile. Nanotechnology strategies, including microparticles, nanostructured lipid carriers, nanoparticles, nanoemulsions, microemulsions, liposomes, phytosomes, niosomes and transferosomes, have been employed for this purpose [79]. The administration of quercetin is typically well tolerated, though minor side effects such as mild headache, nausea and tingling of the extremities may occur with long-term supplementation at 1000 mg/day. Nephrotoxicity has been documented with high intravenous doses in cancer patients [80].

Naringin was chosen as the second flavonoid for MD simulations due to its safety profile and efficacy, as outlined in the scientific opinion issued by the EFSA [43,51]. Moreover, it emerged as one of the highest-ranked phytochemicals, exhibiting a docking score of −10.0 kcal/mol for p38b (ranking 13th) and −9.1 kcal/mol for ERK1 (ranking 51st), with minimal variance compared to the top-ranked natural compounds for p38b (1.9 kcal/mol) and ERK1 (1.4 kcal/mol). It was predicted to affect a total of 16 bacterial strains, with 7 of them associated with osteoporosis. Hesperidin, the final compound chosen for MD simulations, possesses a high safety threshold in food according to the EFSA [50,52,56]. Additionally, it is currently under clinical trials regarding osteopenia and osteoporosis, with ongoing investigations and no definitive conclusions yet reached (ClinicalTrials.gov Identifier: NCT00330096). Furthermore, it stood out as one of the top-ranking phytochemicals, showcasing a docking score of −9.2 kcal/mol for p38b (ranked 46th) and −9.9 kcal/mol for ERK1 (ranked 13th), with negligible differences compared to the highest-ranked natural compounds for p38b (2.7 kcal/mol) and ERK1 (0.6 kcal/mol). Ultimately, it was predicted to affect 12 strains linked to the gut microbiome, of which 5 linked to osteoporosis. A visual examination of the docking poses of these compounds using the PyMOL 2.5.5 software was also conducted to verify their presence within the allosteric binding pocket.

### 2.2. MD Simulations

To assess the binding of the five selected compounds to p38b and ERK1, we conducted 100 ns MD simulations on the protein–ligand complexes using Gromacs 2021 [81]. The top-ranked docking pose of each protein–ligand complex was used as a starting structure. Each simulation was performed in triplicate with identical starting conditions but different initial velocities. The calculation of the ligands’ root-mean-square deviation (RMSD) with respect to the initial frame showed that quercetin (RMSD_quercetin_ = 2.74 ± 0.22 Å—Figure 3A, RMSD_backbone_ = 1.68 ± 0.13 Å—Figure 3B) and naringin (RMSD_naringin_ = 3.00 ± 0.89 Å—Appendix A, RMSD_backbone_ = 1.97 ± 0.32 Å—Appendix A) remained stable inside the allosteric ERK1 binding pocket. On the other hand, salidroside (RMSD = 6.16 ± 1.74 Å, Appendix A), chlorogenic acid (RMSD = 13.92 ± 10.00 Å, Appendix A) and hesperidin (RMSD = 6.20 ± 0.21 Å, Appendix A) demonstrated notable large RMSD values compared to the protein’s backbone RMSD values (2.23 ± 0.16 Å—Appendix A, 2.67 ± 0.82 Å—Appendix A and 2.87 ± 0.35 Å—Appendix A, respectively), highlighting that the phytochemicals are drifting off the binding site of the receptor. The ligand RMSD values for each protein–ligand complex for each replica are listed in Table 2, while the protein backbone RMSD values are listed in Appendix A.

To further study the capability of quercetin and naringin to act as potential inhibitors of ERK1, we computed the binding free energy of the two natural compounds and the co-crystalized ligand (SCH772984) on ERK1 using the MM-PBSA method (Table 3). Quercetin exhibited the most favorable free energy of binding on ERK1 (−25.54 ± 8.13 kJ/mol), closely resembling the binding of the co-crystalized ligand (−33.80 ± 8.95 kJ/mol). Consequently, quercetin demonstrates significant inhibitory potential against ERK1. On the contrary, the binding free energy of naringin on the same protein was positive in all three simulations (+15.20 ± 5.88 kJ/mol), indicating that naringin is not anticipated to bind to the receptor.

To investigate the structural basis of the binding of quercetin with ERK1, we calculated the interactions between the natural compound and the amino acids of the binding site of the receptor, and then we extracted their frequency of appearance using the GetContacts module [82] (Figure 3C). The interactions of SCH772984 with ERK1 were similarly assessed for comparison with the interactions observed with quercetin. A crucial hydrogen bond established by quercetin with ERK1 involves the amino acid ASP184, occurring with a frequency of 99 ± 1%. This interaction has been extensively documented in the previous literature as significant for the adoption of the DFG ‘in’ conformation [41]. Moreover, it forms strong hydrogen bonds with LYS71 (69 ± 29% frequency), GLU88 (32 ± 8% frequency) and LYS131 (90 ± 8% frequency) that are considered as three key interactions contributing to the tight binding of inhibitors [42]. In addition to this, quercetin establishes notable interactions with the allosteric site of ERK1, particularly with TYR53 (96 ± 1% frequency) and MET125 (97 ± 1% frequency). These interactions exhibit comparable frequencies to those observed with SCH772984. Finally, the first cluster representative of the quercetin–ERK1 complex highlighting the most important interactions of quercetin with ERK1 is illustrated in Figure 3D.

Concerning p38b, all phytochemicals displayed low RMSD values (RMSD_naringin_ = 2.40 ± 0.20 Å—Figure 4A, RMSD_backbone_ = 2.76 ± 0.14 Å—Figure 4B, RMSD_chlorogenic acid_ = 2.34 ± 0.41 Å—Appendix A, RMSD_backbone_ = 2.75 ± 0.43 Å—Appendix A, RMSD_quercetin_ = 3.12 ± 0.53 Å—Appendix A, RMSD_backbone_ = 2.89 ± 0.20 Å—Appendix A and RMSD_hesperidin_ = 3.84 ± 0.27 Å—Appendix A, RMSD_backbone_ = 3.10 ± 0.28 Å—Appendix A), indicating their stability within the allosteric binding site. However, salidroside appeared to become solvent-exposed in replica 3 (RMSD_salidroside_ = 5.77 ± 3.22 Å—Appendix A, RMSD_backbone_ = 2.99 ± 0.41 Å—Appendix A). The ligand RMSD values for each protein–ligand complex for each replica are listed in Table 4, while the protein backbone RMSD values are listed in Appendix A.

To delve deeper into the potential inhibitory effects of the four natural compounds on p38b, we employed the MM-PBSA method to compute the binding free energy of naringin, chlorogenic acid, quercetin and hesperidin, along with the co-crystalized ligand (nilotinib), on p38b (Table 5). Naringin demonstrated the most favorable binding free energy on p38b equal to −13.69 ± 6.34 kJ/mol, followed by hesperidin with −5.01 ± 11.12 kJ/mol and chlorogenic acid with −3.41 ± 13.15 kJ/mol. However, none of the three compounds reached the binding free energy of nilotinib (−38.94 ± 30.95 kJ/mol), with two out of three replicas recording values exceeding 50 kJ/mol. Finally, quercetin exhibited negligible binding affinity for p38b, with a value of +1.92 ± 0.82 kJ/mol.

To explore how naringin may bind with p38b on a molecular level, we computed the interactions between the phytochemical and the residues present in the receptor’s binding site as well as their frequency of appearance (Figure 4C). We also calculated the interactions of nilotinib with p38b in a similar manner for comparison reasons. Naringin establishes crucial interactions with the allosteric site of p38b, particularly with PHE169 (30 ± 17% frequency) and ASP168 (99 ± 1% frequency). These interactions exhibit comparable frequencies to those observed with nilotinib. Finally, the first cluster representative of the naringin–p38b complex highlighting the most important interactions of naringin with p38b is illustrated in Figure 4D.

In light of the inhibitory potential exhibited by naringin towards p38b and quercetin towards ERK1, our investigation extended to determine their binding capabilities to additional MAPKs, namely JNK1/2/3, p38c and ERK2. Concerning JNK1, naringin exhibited conformational flexibility within the allosteric pocket, as indicated by an RMSD of 5.51 ± 2.25 Å (Appendix A), contrasting with the stability of the backbone with an RMSD of 2.54 ± 0.07 Å (Appendix A). Conversely, quercetin demonstrated the ability to effectively bind within the same pocket (RMSD_quercetin_ = 3.47 ± 0.56 Å—Appendix A, RMSD_backbone_ = 2.10 ± 0.11 Å—Appendix A). However, both naringin and quercetin displayed an unfavorable free energy of binding to JNK1, with values of +13.88 ± 1.89 kJ/mol and +2.40 ± 7.62 kJ/mol, respectively. In the case of p38c, both naringin and quercetin were found to have drifted away from the allosteric binding site of the host, as indicated by their respective RMSD values of 15.26 ± 7.19 Å (Appendix A) and 9.72 ± 2.07 Å (Appendix A). This contrasts with the stability observed in the protein backbone, with RMSD values of 4.02 ± 0.37 Å (Appendix A) and 4.73 ± 0.53 Å (Appendix A). For ERK2, quercetin became solvent-exposed, with an RMSD of 19.48 ± 13.22 Å (Appendix A), while the protein backbone exhibited a low RMSD value of 1.82 ± 0.12 Å (Appendix A). Conversely, naringin displayed conformational flexibility within the allosteric pocket, with an RMSD of 4.94 ± 1.30 Å (Appendix A), contrasting with the stability observed in the backbone, which had an RMSD of 2.01 ± 0.06 Å (Appendix A). However, naringin exhibited an unfavorable free energy of binding to ERK2, measuring at +20.42 ± 3.53 kJ/mol. The ligand RMSD values for the JNK1–ligand complexes, p38c-ligand complexes and ERK2-ligand complexes for each replica are listed in Appendix A, while the backbone RMSD values for each protein are listed in Appendix A.

Regarding JNK2, both naringin and quercetin demonstrated notably low RMSD values (RMSD_naringin_ = 3.06 ± 0.21 Å—Appendix A and RMSD_quercetin_ = 3.30 ± 1.02 Å—Figure 5A) comparable to those of the protein backbone ones (2.58 ± 0.59 Å—Appendix A and 2.07 ± 0.25 Å—Figure 5B for naringin and quercetin, respectively). The ligand RMSD values for each protein–ligand complex for each replica are listed in Table 6, while the protein backbone RMSD values are listed in Appendix A.

Furthermore, both compounds exhibited favorable binding free energies (Table 7), with naringin at −7.29 ± 9.81 kJ/mol and quercetin at −4.65 ± 4.59 kJ/mol. However, neither of them reached the computed binding free energy of BIRB-796 (−20.75 ± 2.69 kJ/mol).

Finally, we also investigated the structural basis of the binding of quercetin with JNK2 by calculating the interactions between the natural compound and the amino acids of the binding site of the receptor, and then we extracted their frequency of appearance using the GetContacts module (Figure 5C). The interactions of BIRB-796 with JNK2 were similarly assessed for comparison with the interactions observed with quercetin. Quercetin forms crucial interactions with the “back pocket” of JNK2 that is sandwiched between the side chains of LYS55 (98 ± 1% frequency), MET108 (99 ± 1% frequency) and LYS168 (99 ± 1% frequency) [83]. It also establishes lipophilic contacts with LEU110 (66 ± 23% frequency) and MET111 (92 ± 7% frequency). Additionally, it forms two crucial hydrogen bonds with the side chain of LYS73 situated in helix C (99 ± 1% frequency) and ASP169 (99 ± 1% frequency) found in the DFG motif. Finally, it engages in hydrophobic interactions with PHE170 (89 ± 1% frequency) which is located in the activation loop. The first cluster representative of the quercetin–JNK2 complex highlighting the most important interactions of quercetin with JNK2 is illustrated in Figure 5D.

Ultimately, we also explored the inhibitory potential of quercetin and naringin towards JNK3. Both natural compounds exhibited conformational flexibility within the allosteric binding pocket, as indicated by the ligand RMSD values of 5.41 ± 1.21 Å (Figure 6A) for quercetin and 6.06 ± 2.09 Å (Appendix A) for naringin, while the protein backbone demonstrated low RMSD values of 2.45 ± 0.08 Å (Figure 6Β) for quercetin and 2.81 ± 0.35 Å for naringin (Appendix A). The ligand RMSD values for each protein–ligand complex for each replica are listed in Table 8, while the protein backbone RMSD values are listed in Appendix A.

Moreover, naringin showed an unfavorable binding free energy of +13.69 ± 9.49 kJ/mol, whereas quercetin exhibited a slightly negative binding free energy of −0.11 ± 0.62 kJ/mol (Table 9). However, quercetin failed to achieve the computed binding free energy of X3S, which was measured at −12.63 ± 17.21 kJ/mol. Remarkably, there were substantial variations among the three runs for the co-crystalized ligand, with replica 1 showing a positive value of binding free energy at +5.44 ± 2.74 kJ/mol. This highlights the challenge of identifying new inhibitors for this protein.

Finally, we examined the structural mechanism underlying the binding of quercetin to JNK3. This involved calculating the interactions between the natural compound and the amino acids within the receptor’s binding site, followed by determining their frequency of occurrence using the GetContacts module (Figure 6C). To provide a basis for comparison, we also evaluated the interactions of X3S with JNK3 in a similar manner. Quercetin establishes essential interactions with LYS93 (occurring with a frequency of 64 ± 27%) within hydrophobic pocket-I and with ILE70 (frequency: 87 ± 13%), VAL78 (frequency: 43 ± 12%), VAL196 (frequency: 70 ± 11%) and LEY206 (frequency: 81 ± 1%) situated at hydrophobic pocket-II [84]. The first cluster representative of the quercetin–JNK3 complex highlighting the most crucial interactions of the natural compound with JNK3 is depicted in Figure 6D.

### 2.3. In Vitro Assays

To experimentally determine the interaction of the five natural compounds with ERK1 and p38b, the KdELECT technique from Eurofins (https://www.eurofinsdiscovery.com, accessed on 14 March 2024) was performed. After conducting two experimental runs, it was evident that only quercetin exhibited a robust binding affinity towards ERK1, with a measured value of 70 µM. On the other hand, none of the five compounds displayed any binding to p38b (>100 µM). Furthermore, we investigated the potential inhibitory effects of quercetin and naringin on JNK2 and JNK3. Naringin failed to bind to either of these two proteins at concentrations exceeding 100 µM. In contrast, quercetin displayed strong inhibition for both JNK2 (11 μΜ) and JNK3 (3.6 nM). The KdELECT results illustrating the binding of the examined phytochemical to the three target proteins are depicted in Figure 7. The results highlight the substantial potential of quercetin in modulating the activity of kinases relevant to bone metabolism and osteoporosis through its binding to three distinct MAPKs.

The proposed mechanism of quercetin’s action on osteoclast differentiation and activity, impacting bone metabolism, involves its modulation of the NF-κB/MAPK/Akt signaling pathway, as depicted in Figure 8. Upon RANKL binding, RANK triggers the activation of the NF-κB/MAPK/Akt pathway, facilitating NFATc1 self-amplification. NFATc1, in turn, regulates the expression of osteoclast-specific genes such as c-fos, CTSK, Acp5, DC-stamp and V-ATpase-D2. Our findings suggest that quercetin may inhibit osteoclastogenesis by potentially interrupting this cascade, with a particular focus on ERK1, JNK2 and JNK3 proteins. While further experimental validation is needed to substantiate this hypothesis, these findings emphasize quercetin’s potential importance in osteoporosis and lay the groundwork for the creation of pioneering treatment approaches for the condition.

## 3. Materials and Methods

### 3.1. The Selection of the Most Promising Natural Compounds for MD Simulations

To examine the inhibitory potential of several classes of natural compounds to the p38, ERK and JNK isoforms, an initial set of 187 phytochemicals was collected from CNatural, an online platform developed by Cloudpharm (http://cnatural.eu/home/, accessed on 14 March 2024) containing more than 180,000 natural products collected from four databases (ChEMBL, ChEBI, NPASS and CTD). The compounds under study were selected based on their reported association in osteoporosis from literature data.

Selecting the receptor crystal structure greatly influences the successful prediction of the stability of a protein–ligand complex [85]. Therefore, the crystal structures of the p38, ERK and JNK isoforms were determined based on the following criteria: (1) proteins ought to be of human origin, (2) resolution lower than 2.8 Å, (3) the presence of an allosteric co-crystalized ligand and (4) no mutations on the amino acids of the binding sites defined by the corresponding ligand (residues within 5 Å distance from the ligand). The atomistic models with PDB IDs are as follows: 3GP0 [86], 1CM8 [87], 4QTB [41], 4QTA [41], 3ELJ [88], 3NPC [83] and 7KSI [84] and met the above criteria for the p38b, p38c, ERK1, ERK2, JNK1, JNK2 and JNK3 proteins.

The selected crystal structures along with the 187 natural compounds were used for the docking calculations. AutoDock Tools were used for the preparation of the proteins and the compounds [89], while AutoDock Vina 1.1.2 [36] was employed to dock the phytochemicals into the receptors. The binding site was defined from the center of the co-crystalized ligands of each PDB structure, using grid cubic boxes with dimensions of 20 × 20 × 20 Å. Proteins were held rigid, while the natural compounds were allowed to be flexible during docking simulations. For each compound, 10 poses were generated, with the exhaustiveness parameter set to 20. To validate our docking protocol, we assessed the ability of AutoDock Vina to correctly dock nilotinib and SCH772984 in the allosteric binding pockets of p38b and ERK1, respectively.

Moreover, the impact of the 187 natural compounds on the growth of 40 gut bacterial strains was studied utilizing an ML algorithm developed by McCoubrey et al. [70]. To the best of our knowledge at the time of conducting our ML study (early 2023), there were no other pertinent ML-based methods available for delineating the impact of drugs on the growth of bacterial strains within the human gut microbiome. Despite the potential illumination such studies could provide regarding the impact of drugs on the microbiome and the subsequent opportunity for enhancing pharmaceutical treatments, there remained a deficiency in integrating chemical and microbiological knowledge with a computational, data-driven, systems approach. According to this algorithm, 1613 molecular descriptors were generated for each compound based on their simplified molecular-input line-entry system (SMILES) using the Mordred package [90]. Based on these features, the best performing ML model from McCoubrey et al. (extra trees classifier with performance metrics of AUROC: 0.86 ± 0.01, weighted recall: 0.59 ± 0.06, weighted precision: 0.80 ± 0.05 and weighted f1: 0.67 ± 0.04) was employed to detect the gut bacterial effects of the 187 natural compounds.

### 3.2. MD Simulations

The most promising compounds were then subjected to 100 ns MD simulations in triplicate using the Gromacs 2021 [81] and CHARMM36 (July 2021 version) force field [91]. As starting structures for the MD simulations, we used the top docked pose from the p38b and ERK1 AutoDock Vina docking calculations. Missing residues were modeled by utilizing the CHARMM-GUI web-server [92]. The force field parameters for small molecules were generated using CGenFF [93], while the TIP3P model [94] was used to model the water molecules. Simulation boxes were created with in a cubic box ensuring a minimum distance of 15 Å between each complex atom and the edge of the periodic box.. The protein–ligand complexes were solvated, and then sodium and chloride ions were added to neutralize the charge of the system. All systems were energy-minimized for 50,000 steps, using the steepest decent algorithm [95,96]. Then, they were subjected to NPT equilibration for 10 ns with a 2 fs timestep, using positional restraints for the ligands. A Berendsen barostat [97] was employed to maintain a target pressure of 1 bar with a time constant of 2 ps and a compressibility of 4.5 × 10^−5^ bar^−1^, and the temperature control was kept at 300 K using the V-rescale thermostat [98] with a coupling constant of 0.1 ps. Once the systems were equilibrated, 100 ns MD simulations were performed in the NPT ensemble with the atomic coordinates saved every 100 ps. Covalent bonds with hydrogen atoms were constrained using the LINear Constraint Solver (LINCS) [99] algorithm allowing for a 2 fs timestep. A Parrinello–Rahman barostat [100] was used to keep pressure at 1 bar, with a time constant of 2 ps and a compressibility of 4.5 × 10^−5^ bar^−1^, and a V-rescale thermostat was used to keep temperature at 300 K, with a coupling constant of 0.1 ps. Long-range electrostatic interactions were treated using the particle mesh Ewald method [101] with a maximum grid spacing of 1.2 Å. Non-bonded interactions were calculated with a cutoff of 12 Å and a switching distance of 10 Å.

The MD trajectories were postprocessed and analyzed using the standard GROMACS tool (gmx trjconv) for centering and fitting based on the protein atoms. The ligand’s RMSD and protein’s backbone RMSD as a function of time were calculated through the MDAnalysis Python library [102,103] using the starting frame as a reference structure. The gmx_MMPBSA module [104] was employed to estimate the binding free energy of the protein–ligand complexes using MM-PBSA calculations [105]. A total of 51 snapshots (equally spaced in the last 50 ns of the MD simulations) were used for the calculation of the binding free energy. The GetContacts module [82] was employed to calculate the contacts of the co-crystalized ligands and the natural products with the protein residues during the MD simulations. The cutoff ranges for hydrogen bonds were set to 3.5 Å and 70° angles, for hydrophobic interactions, 6 Å and 45° angles and for salt bridges, 4 Å. Finally, to obtain the most representative structure of the protein–ligand complexes, we clustered the conformations using the gromos [106] algorithm of the gmx_cluster routine (GROMACS 2021.5). The RMSD of the non-hydrogen atoms of each protein–ligand complex was used as the clustering criterion, and a cutoff value of 1 Å was chosen in order to obtain balanced cluster sizes. The central structure of the most populated cluster was picked in order to acquire the most representative structure of each system.

### 3.3. In Vitro Assays

To experimentally determine the binding of the five most promising compounds obtained from the in silico studies to four proteins of the MAPK pathway (p38b, ERK1, JNK2 and JNK3), the KdELECT technique from Eurofins (https://www.eurofinsdiscovery.com, accessed on 14 March 2024) was employed. It is based on the KINOMEscan™ assay developed by DiscoverX, which is one of the most comprehensive high-throughput systems for screening compounds against large numbers of human kinases. KINOMEscan™ operates on a competition binding assay, quantifying a compound’s ability to compete with an immobilized, active site-directed ligand. This assay integrates three key components: DNA-tagged kinase, the immobilized ligand and the test compound. The competitive interaction between the test compound and the immobilized ligand is evaluated through a quantitative PCR analysis of the DNA tag [107].

## 4. Conclusions

In this study, we demonstrated the potential of natural compounds, specifically quercetin, to act as inhibitors of the MAPK pathway, which plays a critical role in osteoclast differentiation and activity, potentially influencing bone metabolism. We developed a novel workflow for screening inhibitors of the MAPK pathway, incorporating thorough computational analyses such as molecular docking, molecular dynamics simulations, machine learning and binding free energy calculations of natural compounds. Within this framework, among five selective compounds, quercetin exhibited significant binding affinity and inhibitory potential towards ERK1 and JNK isoforms. Moreover, in vitro assays confirmed quercetin’s binding affinity towards ERK1, JNK2 and JNK3, highlighting its potential role as a multi-target inhibitor with potential therapeutic benefits for osteoporosis management. Although further experimental validation is necessary, these findings may open new avenues for osteoporosis treatment.

## Figures and Tables

**Figure 1 ijms-25-05047-f001:**
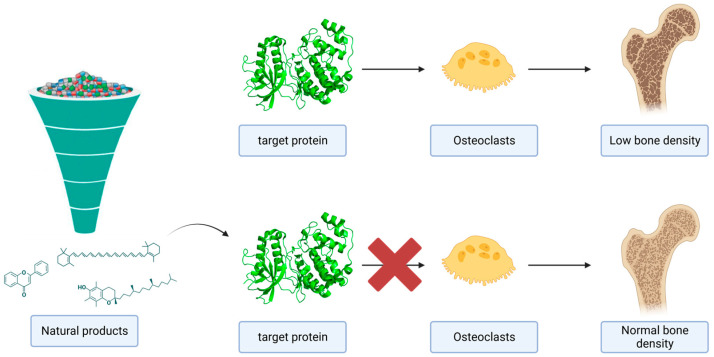
A graphical summary of the role of MAPKs in bone metabolism. The potential inhibition of MAPKs by natural products could suppress the differentiation and activation of the osteoclasts.

**Figure 2 ijms-25-05047-f002:**
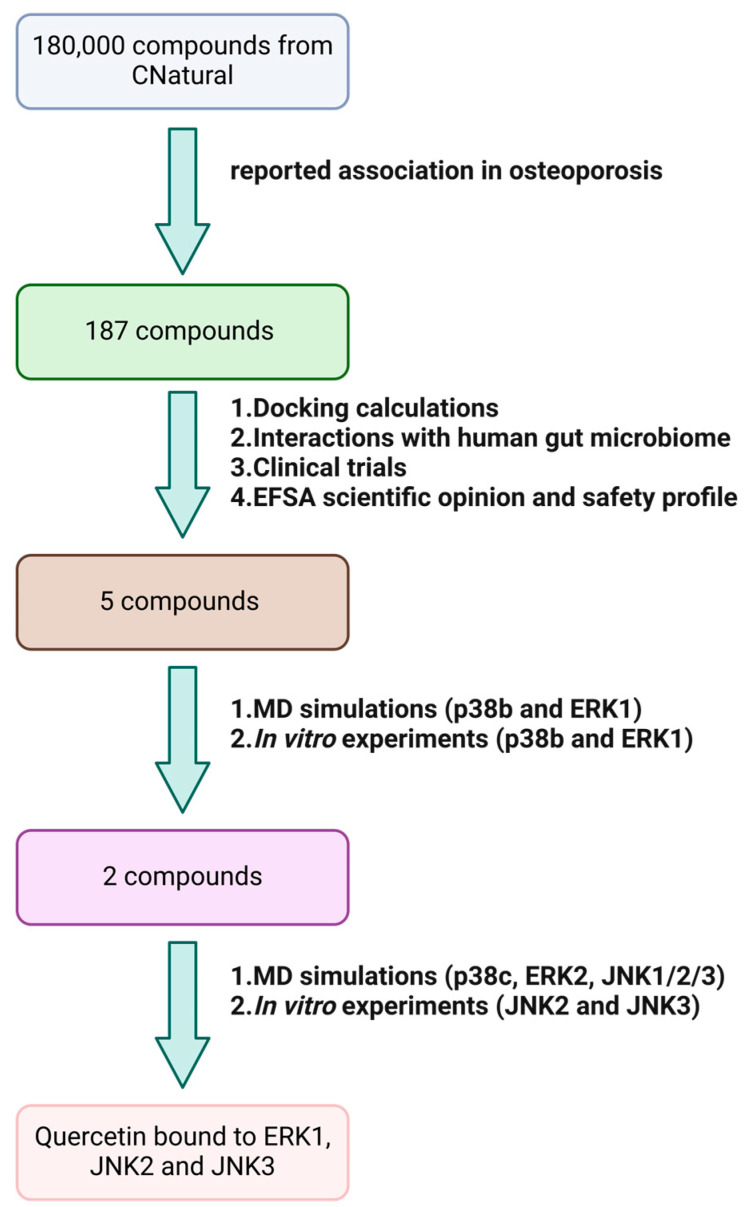
An overview of the workflow implemented for the discovery of natural compounds that have the potential to inhibit MAPKs. The procedure starts with a virtual library enumeration of 180,000 compounds that are filtered based on their reported association with osteoporosis (187 compounds). It then continues with docking calculations, prediction of the interactions of the compounds with the human gut microbiome and assessment of potential clinical trials that have participated and respective EFSA scientific opinions and safety profiles to produce a subset of five ligands in allosteric binding mode. Then, MD simulations are employed on p38b and ERK1 to retain only 2 stably bound molecules, which were subsequently validated in vitro. These compounds undergo testing on additional JNK, p38 and ERK isoforms, followed by a second round of KdELECT assays to identify the natural compound with the potential to modulate the activity of MAPKs. Quercetin emerges as an inhibitor of ERK1, JNK2 and JNK3, suggesting its promise in addressing osteoporosis.

**Figure 3 ijms-25-05047-f003:**
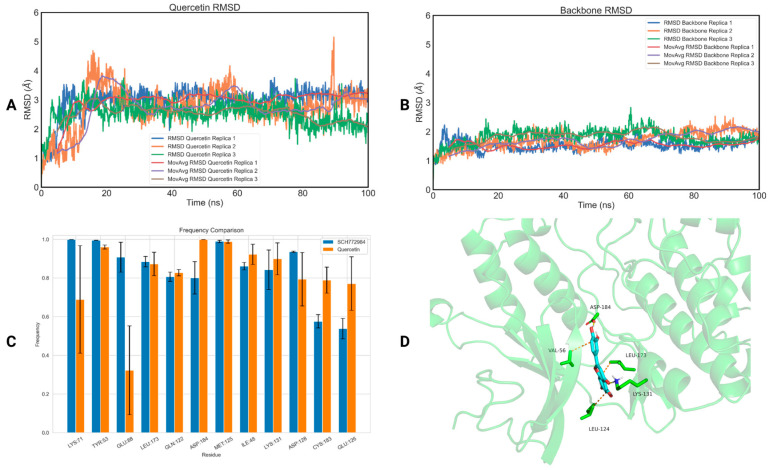
(**A**) RMSD of the three replicas of quercetin bound to ERK1 as a function of time; (**B**) RMSD of the three replicas of the backbone of ERK1 as a function of time; (**C**) a diagram of the interactions of quercetin with ERK1 (orange), in comparison with the interactions of SCH772984 with ERK1 (blue); (**D**) the first cluster representative of quercetin bound to ERK1.

**Figure 4 ijms-25-05047-f004:**
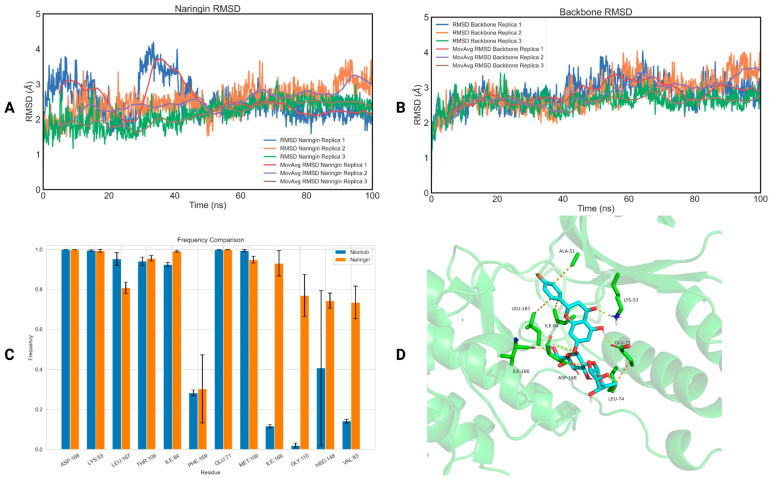
(**A**) RMSD of the three replicas of naringin bound to p38b as a function of time; (**B**) RMSD of the three replicas of the backbone of p38b as a function of time; (**C**) a diagram of the interactions of naringin with p38b (orange), in comparison with the interactions of nilotinib with p38b (blue); (**D**) the first cluster representative of naringin bound to p38b.

**Figure 5 ijms-25-05047-f005:**
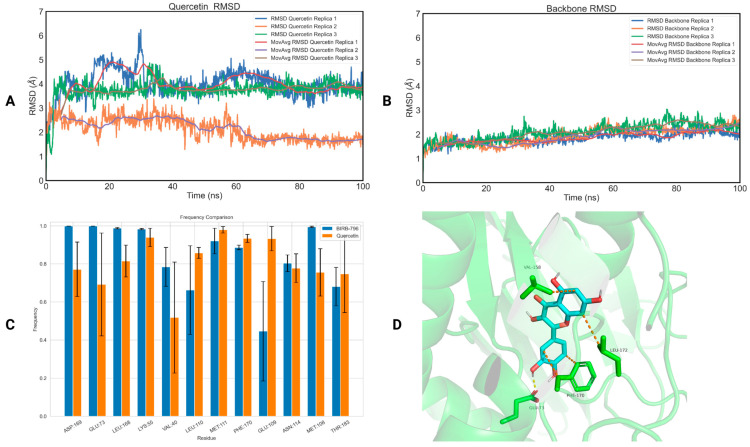
(**A**) RMSD of the three replicas of quercetin bound to JNK2 as a function of time; (**B**) RMSD of the three replicas of the backbone of JNK2 as a function of time; (**C**) a diagram of the interactions of quercetin with JNK2 (orange), in comparison with the interactions of BIRB-796 with JNK2 (blue); (**D**) the first cluster representative of quercetin bound to JNK2.

**Figure 6 ijms-25-05047-f006:**
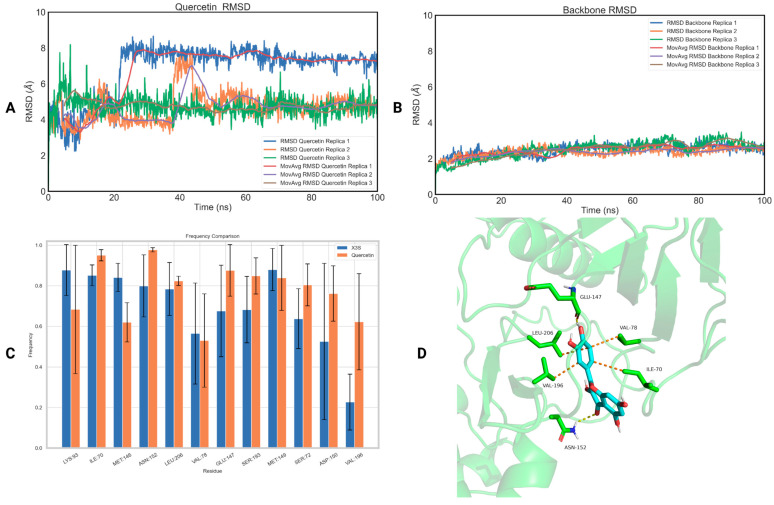
(**A**) RMSD of the three replicas of quercetin bound to JNK3 as a function of time; (**B**) RMSD of the three replicas of the backbone of JNK3 as a function of time; (**C**) a diagram of the interactions of quercetin with JNK3 (orange), in comparison with the interactions of X3S with JNK3 (blue); (**D**) the first cluster representative of quercetin bound to JNK3.

**Figure 7 ijms-25-05047-f007:**
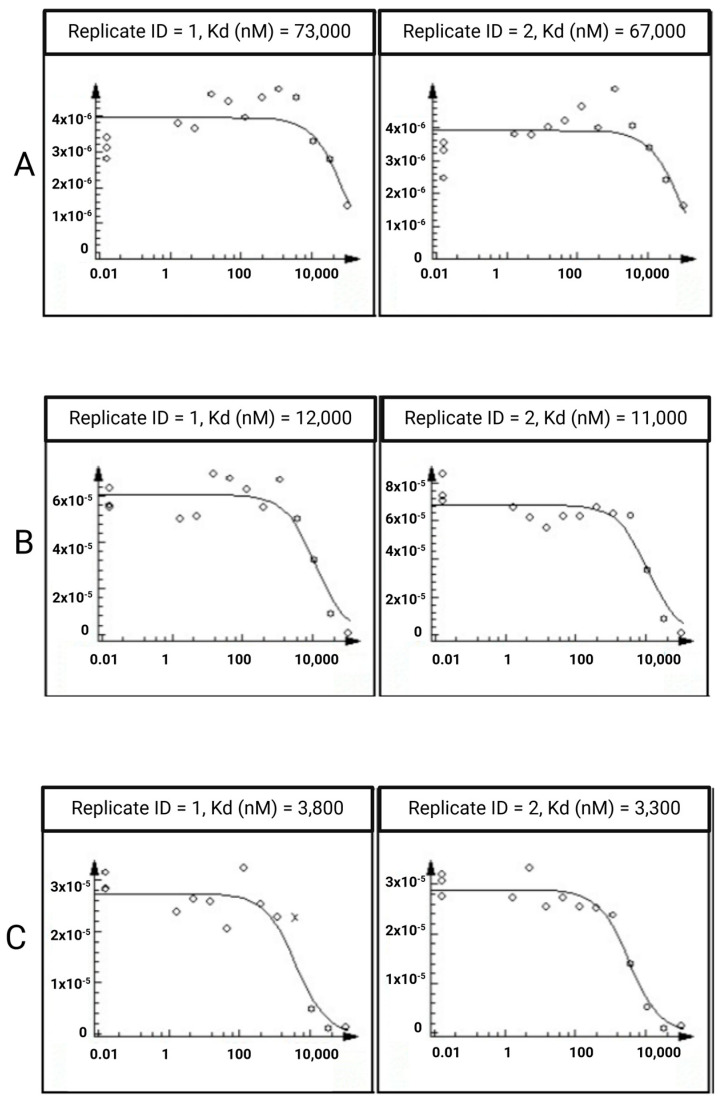
The amount of (**A**) ERK1, (**B**) JNK2 and (**C**) JNK3 measured by qPCR (Signal; *y*-axis) is plotted against the corresponding quercetin concentration in nM in log10 scale (*x*-axis).

**Figure 8 ijms-25-05047-f008:**
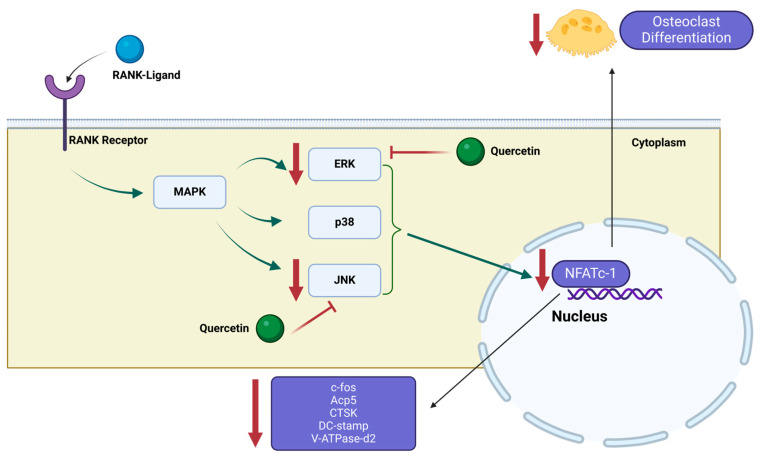
Proposed mechanism for quercetin’s inhibition of osteoclastogenesis.

**Table 1 ijms-25-05047-t001:** The compounds selected for MD simulations, their predicted collective impact on bacterial strains and the specific subset of strains demonstrating either positive or negative effects on osteoporosis.

Natural Compound	2D Structure	Total Number of Affected Bacterial Strains	Number of Affected Strains Positively Correlated with Osteoporosis	Number of Affected Strains Negatively Correlated with Osteoporosis
Quercetin	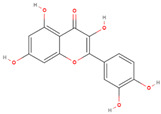	22	8	7
Chlorogenic acid	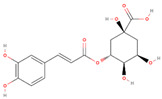	18	6	5
Naringin	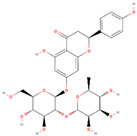	16	4	3
Hesperidin	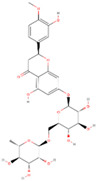	12	4	1
Salidroside	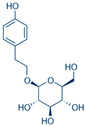	6	2	1

**Table 2 ijms-25-05047-t002:** Average values of RMSD and standard deviation of each ERK1–ligand complex were calculated over 100 ns MD simulations.

Replica ID	RMSD Quercetin (Å)	RMSD Naringin (Å)	RMSD Salidroside (Å)	RMSD Chlorogenic Acid (Å)	RMSD Hesperidin (Å)
1	2.96 ± 0.49	2.47 ± 0.57	8.21 ± 5.52	11.50 ± 1.87	6.15 ± 1.25
2	2.74 ± 0.71	2.49 ± 0.49	3.95 ± 1.29	5.35 ± 1.49	6.03 ± 1.09
3	2.52 ± 0.40	4.03 ± 1.49	6.33 ± 3.00	24.90 ± 22.93	6.43 ± 1.27
Average RMSD	2.74 ± 0.22	3.00 ± 0.89	6.16 ± 1.74	13.92 ± 10.00	6.20 ± 0.21

**Table 3 ijms-25-05047-t003:** Binding free energies (kJ/mol) for the ERK1–ligand complexes as obtained by the MM/PBSA calculations.

Replica ID	Free Energy of Binding Quercetin (kJ/mol)	Free Energy of Binding Naringin (kJ/mol)	Free Energy of Binding SCH772984 (kJ/mol)
1	−16.84 ± 1.21	+17.91 ± 5.22	−42.65 ± 2.52
2	−26.85 ± 2.66	+8.45 ± 7.42	−24.75 ± 0.93
3	−32.94 ± 5.31	+19.24 ± 3.53	−34.01 ± 2.32
Average free energy of binding	−25.54 ± 8.13	+15.20 ± 5.88	−33.80 ± 8.95

**Table 4 ijms-25-05047-t004:** Average values of RMSD and standard deviation of each p38b–ligand complex were calculated over 100 ns MD simulations.

Replica ID	RMSD Quercetin (Å)	RMSD Naringin (Å)	RMSD Salidroside (Å)	RMSD Chlorogenic Acid (Å)	RMSD Hesperidin (Å)
1	3.40 ± 0.89	2.53 ± 0.53	3.74 ± 0.89	2.53 ± 0.73	4.06 ± 0.96
2	3.44 ± 0.70	2.51 ± 0.41	3.25 ± 0.70	1.87 ± 0.33	3.92 ± 0.54
3	2.51 ± 0.51	2.17 ± 0.37	10.31 ± 0.51	2.62 ± 0.47	3.53 ± 0.49
Average RMSD	3.12 ± 0.53	2.40 ± 0.20	5.77 ± 3.22	2.34 ± 0.41	3.84 ± 0.27

**Table 5 ijms-25-05047-t005:** Binding free energies (kJ/mol) for the p38b–ligand complexes as obtained by the MM/PBSA calculations.

Replica ID	Free Energy of Binding Quercetin (kJ/mol)	Free Energy of Binding Naringin (kJ/mol)	Free Energy of Binding Hesperidin (kJ/mol)	Free Energy of Binding Chlorogenic Acid (kJ/mol)	Free Energy of Binding Nilotinib (kJ/mol)
1	+2.79 ± 6.99	−15.87 ± 5.11	−0.57 ± 11.24	−18.14 ± 9.92	−3.85 ± 2.73
2	+1.17 ± 6.11	−18.66 ± 6.50	−17.67 ± 9.53	+7.15 ± 9.15	−62.35 ± 1.54
3	+1.80 ± 4.87	−6.55 ± 4.14	+3.20 ± 5.62	+0.77 ± 5.88	−50.62 ± 3.52
Average free energy of binding	+1.92 ± 0.82	−13.69 ± 6.34	−5.01 ± 11.12	−3.41 ± 13.15	−38.94 ± 30.95

**Table 6 ijms-25-05047-t006:** Average values of RMSD and standard deviation of each JNK2–ligand complex were calculated over 100 ns MD simulations.

Replica ID	RMSD Quercetin (Å)	RMSD Naringin (Å)
1	2.36 ± 0.41	2.96 ± 0.20
2	1.90 ± 0.24	2.88 ± 0.42
3	1.95 ± 0.25	3.35 ± 0.30
Average RMSD	2.07 ± 0.25	3.06 ± 0.21

**Table 7 ijms-25-05047-t007:** Binding free energies (kJ/mol) for the JNK2–ligand complexes as obtained by the MM/PBSA calculations.

Replica ID	Free Energy of Binding Quercetin (kJ/mol)	Free Energy of Binding Naringin (kJ/mol)	Free Energy of Binding BIRB-796 (kJ/mol)
1	−6.54 ± 3.44	−16.97 ± 6.49	−23.51 ± 2.37
2	−7.99 ± 1.88	−11.07 ± 8.43	−21.65 ± 2.42
3	+0.59 ± 2.76	+6.17 ± 7.51	−17.10 ± 1.79
Average free energy of binding	−4.65 ± 4.59	−7.29 ± 9.81	−20.75 ± 2.69

**Table 8 ijms-25-05047-t008:** Average values of RMSD and standard deviation of each JNK3–ligand complex were calculated over 100 ns MD simulations.

Replica ID	RMSD Quercetin (Å)	RMSD Naringin (Å)
1	6.82 ± 1.44	3.96 ± 0.81
2	4.69 ± 0.85	5.30 ± 1.42
3	4.74 ± 0.56	8.91 ± 1.84
Average RMSD	5.41 ± 1.21	6.06 ± 2.09

**Table 9 ijms-25-05047-t009:** Binding free energies (kJ/mol) for the JNK3–ligand complexes as obtained by the MM/PBSA calculations.

Replica ID	Free Energy of Binding Quercetin (kJ/mol)	Free Energy of Binding Naringin (kJ/mol)	Free Energy of Binding X3S (kJ/mol)
1	−0.52 ± 5.08	+4.41 ± 4.58	+5.44 ± 2.37
2	+0.76 ± 3.51	+9.94 ± 6.84	−7.54 ± 5.46
3	−0.57 ± 5.35	+26.72 ± 7.22	−35.79 ± 2.41
Average free energy of binding	−0.11 ± 0.62	+13.69 ± 9.49	−12.63 ± 17.21

## Data Availability

The raw data supporting the conclusions of this article will be made available by the authors on request.

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
