# Peer review of "Natural Compounds for Bone Remodeling: A Computational and Experimental Approach Targeting Bone Metabolism-Related Proteins"

_ijms, 2024, doi:10.3390/ijms25095047_

Round 1
Reviewer 1 Report
Comments and Suggestions for Authors
Loukas et al. applied a combination of machine learning, molecular docking, and simulation studies to define the efficacy of natural compounds in modulating molecular targets relevant to osteoporosis. Then, they validated the argument with in vitro studies. The topic is original and addresses gaps in bone hemostasis by focusing on the MAPK pathway and the gut microbiome. As the study applies virtual screening of compounds using computational studies before in vitro analyses, it prevents serendipitous non-required in vitro analyses observed in other publications. The conclusion is consistent with the evidence and arguments, and the references are appropriate. The authors should address the following comments as well
1- The authors should respond that compared to other ML-based approaches to define the effect of drugs on the gut microbiome, why they used the McCoubrey method, and describe the limitations of these programs
2- The most optimal candidate (quercetin) should be described in the discussion pharmaceutically regarding pharmacokinetic, pharmacodynamic, side effects and if there are dosage forms available
3- The authors should draw detailed signaling pathways with appropriate schematics
Reviewer 2 Report
Comments and Suggestions for Authors
The manuscript presents a comprehensive study searching for inhibitors of the MAPKs pathway. Important feature of the study is intelligent combination of suitable in silico approaches and their combination with in vitro tests. I would recommend its publication after reflecting some minor issues:
1.I would recommend modification of the title to reflect the combination of in silico/in vitro approaches in this work.
2. More information is needed to describe the results of the prediction of impact of the 187 natural compounds on the growth of 40 gut bacterial strains using ML models.
3. It is not clear what threshold of docking scores has been used for compounds selection in the screening based on docking simulations.
4. It is not clear how do you validate the docking protocol when dock the compounds in the allosteric binding region. Re-docking of the allosteric cocrystalized ligand may be a solution.
5.In my oppinion one of the achievements of this study is the development of a workflow to screen for inhibitors of the MAPKs pathway and this should be stressed in the conclusion.
6.Some typos/stylistic notes:
- examinatino-> examination
- "To experimentally determine the interaction between the five natural compounds with..." -> "To experimentally determine the interaction OF the five natural compounds with...".
Round 2
Reviewer 1 Report
Comments and Suggestions for Authors
The manuscript is improved after revision.